# Clinical and Genetic Characteristics of Calvarial Doughnut Lesions with Bone Fragility in Three Families with a Reccurent *SGMS2* Gene Variant

**DOI:** 10.3390/ijms24098021

**Published:** 2023-04-28

**Authors:** Elena Merkuryeva, Tatiana Markova, Anton Tyurin, Diana Valeeva, Vladimir Kenis, Maria Sumina, Igor Sorokin, Olga Shchagina, Mikhail Skoblov, Maria Nefedova, Rita Khusainova, Ekaterina Zakharova, Elena Dadali, Sergey Kutsev

**Affiliations:** 1Research Centre for Medical Genetics, 115522 Moscow, Russia; elena.merkureva@gmail.com (E.M.);; 2Internal Medicine Department, Bashkir State Medical University, 450008 Ufa, Russia; 3The Turner Scientific Research Institute for Children’s Orthopedics, 196603 Saint Petersburg, Russia; 4State Healthcare Institution of Sverdlovsk Region “Clinical and Diagnostic Center “Mother’s and Child Health Protection”, 620067 Ekaterinburg, Russia; 5Faculty of Dentistry, A.I. Yevdokimov Moscow State University of Medicine and Dentistry, 127473 Moscow, Russia; 6Independent Clinical Bioinformatics Laboratory, 123181 Moscow, Russia; 7Laboratory of Human Molecular Genetics, Institute of Biochemistry and Genetics, 450000 Ufa, Russia; 8Healthy Longevity Center, Ufa University of Science and Technology, 450008 Ufa, Russia; 9Medical Genetics Department, Bashkir State Medical University, 450008 Ufa, Russia

**Keywords:** *SGMS2* gene, bone fractures, low bone mineral density

## Abstract

Calvarial doughnut lesions (CDL) with bone fragility with or without spondylometaphyseal dysplasia (MIM: #126550) is a rare autosomal dominant skeletal disorder characterized by low bone mineral density, spinal and peripheral fractures, and specific sclerotic lesions of the cranial bones. In the current classification of skeletal disorders, the disease is included in the group of bone fragility disorders along with osteogenesis imperfecta. The disease is caused by pathogenic variants in the *SGMS2* gene, the protein product of which is sphingomyelin synthase 2, which primarily contributes to sphingomyelin (SM) synthesis—the main lipid component of the plasma membrane essential for bone mineralization. To date, 15 patients from eight families with CDL with bone fragility have been described in the literature, and a recurrent variant c.148C>T (p.Arg50Ter) in the *SGMS2* gene has been identified, which was found in patients from six families. We diagnosed the disease in 11 more patients from three unrelated families, caused by the same heterozygous nonsense variant c.148C>T (p.Arg50Ter) in the *SGMS2* gene. Our results show wide interfamilial and intrafamilial phenotypic variability in patients with a detected recurrent variant in the *SGMS2* gene, the presence of which must be taken into consideration in the diagnosis of the disease. The primary analysis of this variant will contribute to optimal molecular genetic diagnostics, which can reduce diagnostic costs and time.

## 1. Introduction

CDL with bone fragility with or without spondylometaphyseal dysplasia (MIM: #126550) is an autosomal dominant skeletal disorder characterized by an abnormal bone structure with cortical thinning and low mineral density. The typical symptoms of the disease include multiple spontaneous fractures of the spine and long bones and sclerotic doughnut-shaped lesions in the cranial bones. Clinical features of the disease were first described in 1969 by Keats and Holt [1]. In the following years, a number of authors also observed patients with similar clinical manifestations [2,3]. However, the etiological factor was only identified in 2019 as a result of a multicenter study of six unrelated families from six countries with 13 affected members. The authors showed that the *SGMS2* gene mapped on the long arm of chromosome 4 is responsible for the occurrence of the disease [4]. The protein product of this gene is sphingomyelin synthase 2, which is involved in the synthesis of SM, the major plasma membrane lipid [5,6]. It has been shown that decreased enzyme activity leads to defective bone mineralization. Four unrelated families with 10 affected members shared the same nonsense variant c.148C>T (p.Arg50Ter) in exon 2 of the *SGMS2* gene. It has thereafter been reported that about two additional families had this variant, confirming it to be the “hot spot” in the *SGMS2* gene [7]. It has been demonstrated that the presence of a nonsense variant results in the production of a catalytically inactive enzyme.

On the contrary, in two families with CDL caused by missense variants (p.Ile62Ser and p.Met64Arg) of the *SGMS2* gene, the change in SM metabolism is associated with impaired export of a functional enzyme from the endoplasmic reticulum. It is assumed that there is a significant variety in the severity of clinical manifestations of the disease in patients with a recurrent nonsense variant and the two missense substitutional variants described to date. Thus, two patients with missense variants had a more severe presentation with neonatal fractures, disproportionate dwarfism, and signs of spondylometaphyseal dysplasia revealed on X-ray examination [4]. On the other hand, in patients with nonsense mutations, the presence of osteoporosis in childhood with normal growth rates has been described.

Due to the rare occurrence of this disease, we present a description of the clinical and radiological features of patients with CDL with bone fragility caused by a pathogenic nonsense variant in the *SGMS2* gene: c.148C>T (p.Arg50Ter).

## 2. Results

The clinical and radiological features of 11 patients (6 males and 5 females) from three unrelated families aged from 6 to 56 years old with CDL and bone fragility were studied. In two families, the segregation occurred within three generations, while in the third family, the segregation occurred within two generations. All affected members across the three families had a nonsense variant in the *SGMS2* gene: c.148C>T (p.Arg50Ter). Pedigree charts of the families are shown in Figure 1.

The main clinical manifestations in the examined patients included multiple fractures, low bone mineral density (BMD) revealed on densitometry, and the presence of palpable indurations of the skull bones. Patient data, including demographics, clinical and radiological data, and age at the time of treatment initiation, are summarized in Table 1.

### 2.1. Family 1

Family 1 is Belarusian and has five affected members in three generations. Proband (III-4) is a 6-year-old boy who started to complain (pointed to where it hurt) of lower back pain from the age of 1 year 7 months, according to his parents. This pain usually lasted from 2 weeks to 1 month and limited the patient’s physical activity, as he was afraid to walk the stairs. Taking into account the increasing lower back pain, an MRI of the thoracolumbar spine was performed at the age of 4 years, which revealed signs of compression fractures of the C7, Th 2-8 vertebral bodies (Figure 2).

According to the results of the X-ray densitometry performed at the age of 4 years, the BMD of the lumbar spine was 0.289 g/cm^2^, Z-criterion was −3.2. The laboratory study of calcium phosphate metabolism revealed an increased level of alkaline phosphatase activity and decreased level of 25-hydroxyvitamin D (25(OH)D). Laboratory tests showed the following results: ionized calcium—1.23 mmol/L (referent values: 1.03–1.23 mmol/L), alkaline phosphatase—492 U/l (referent values: 156–369), 25-hydroxyvitamin D [25(OH)D]—28 ng/ml (referent values: 30–100 ng/ml), nonorganic phosphorus—1.55 mmol/L (referent values: 1.45–1.78 mmol/L). During the patient’s observation, there was a tendency toward an increase in the pain syndrome in the back area (pain occurred during coughing, sneezing, and sudden movements), as well as a decrease in BMD values in the lumbar spine according to densitometry data (BMD = 0.259 g/cm^2^, Z-score: −4.2 SD), which served as a medical indication for initiating treatment with bisphosphonate-based medications (pamidronate acid at a dose of 1 mg/kg/day for 3 consecutive days, courses every 4 months). After two courses of treatment, a positive trend was observed: back pain was relieved, and BMD values increased (Table 1).

Control X-ray densitometry at the age of 6 showed positive dynamics: the BMD of the lumbar spine was 0.357 g/cm2, Z-criterion was −1.9. An X-ray examination of the spine at the age of 6 years revealed diffuse rarefaction of the bone pattern of the vertebrae and a biconcave shape of the thoracic vertebral bodies (Figure 2).

The 32-year-old proband’s father had more than 40 spontaneous fractures that occurred at the age of 10 to 18 years, mainly of the bones of the wrist, feet, and radius bones of both upper extremities. At the age of 15, as a result of a car accident, the proband’s father sustained fractures of the femur, pelvic bones, ribs, and skull. The proband’s paternal uncle had more than 20 fractures of the bones of the wrists and feet during his life. From the age of 5, the proband’s grandmother sustained 13 fractures: the radius, ulna, humerus, and bones of the wrists. The 7-year-old proband’s cousin, as well as the 6-year-old proband himself, has not yet sustained peripheral bone fractures.

Multiple sclerotic lesions of the skull bones were found in the proband’s father, grandmother, and paternal uncle, confirmed by X-ray and CT examination of the skull. On the contrary, the physical examination of a cousin of a 7-year-old proband revealed only three elevated skull seals in the frontal and parietal–occipital regions, which were presented on the CT scan in the form of three rounded foci of osteosclerosis in the frontal and parietal bones with a thin sclerotic rim along the contour and with vacuum in the central sections in the shape of a doughnut. According to the results of a CT scan, by the age of 6 years, the proband did not have any sclerotic lesions of the skull.

An X-ray examination of the wrist bones, performed due to short stature (−1.9 standard deviations) in the cousin of the F1 III-1 proband, revealed relative shortening of the fourth and fifth metacarpals (Figure 3).

In addition, since adolescence, the father, uncle, and grandmother of the proband had transient recurrent facial nerve paralysis, affecting both the upper and lower parts of the face. These episodes lasted for month and resolved on their own without treatment.

### 2.2. Family 2

Family 2 is Russian and has four affected members in three generations. Proband (III-2) is an 11-year-old boy with severe primary osteoporosis in childhood and a history of four low-energy peripheral fractures, starting at the age of 1 year 6 months when the first fracture of the right femur occurred. Later on, at the age of 6 and 8 years, the proband sustained fractures of the wrist and foot bones, and at the age of 10 years—a fracture of both bones of the right forearm. At the age of 10 years, after active physical activity (jumping in the street), the child developed severe back pain, which an X-ray examination of the spine revealed were multiple compression fractures of the bodies of Th 3–9 vertebrae and a wedge-shaped deformity of Th 5–7, 9 vertebrae.

According to the results of the X-ray densitometry performed at the age of 11 years, the BMD of the lumbar spine was 0.289 g/cm^2^, Z-criterion was −3.2, which corresponded to osteoporosis. Laboratory tests showed the following results: ionized calcium—2,3 mmol/L (referent values: 1.03–1.23 mmol/L), alkaline phosphatase—182 U/l (referent values: 156–369), 25-hydroxyvitamin D (25(OH)D)—22.76 ng/ml (referent values: 30–100 ng/ml), nonorganic phosphorus—1.45 mmol/L (referent values: 1.45–1.78 mmol/L). At the age of 11 years, therapy with bisphosphonates was initiated. Less than 6 months have elapsed since the initiation of treatment, and at present, it is difficult to assess the efficacy of the treatment.

From the medical history, it is known that the proband’s mother had the first fracture of the right forearm bones at the age of 6, then at the age of 8 years. At the age of 13, a fracture of the finger phalanx of the right hand occurred. At the age of 29, the proband’s mother sustained a fracture of the rib during a fall, and at the age of 30—a fracture of the left talus and right patella. Since the age of 6, the grandmother of the proband had multiple fractures of the ankle joint (about 10). The proband’s uncle sustained the first fracture of the bones of the left leg at the age of 6. From the age of 7 to 10 years, six fractures of the bones of the left and right forearms occurred. At the age of 10, open wire osteosynthesis of the bones of the right forearm was performed.

According to the results of the densitometry, all affected family members were diagnosed with severe osteoporosis, and an X-ray examination of the spine revealed signs of compression fractures of the vertebral bodies (Figure 2).

Examination of an 11-year-old proband and his uncle of the same age revealed bony prominences of the calvaria in the forehead area 3 × 3 cm in size. In the mother and grandmother of the proband, the same formations were found in the forehead area 5 × 5 cm in size and at the back of the head 3 × 3 cm in size (Figure 4). Sclerotic lesions of the skull bones were found in all family members during X-ray and CT examination of the skull (Figure 5).

An objective examination of the proband’s grandmother and uncle revealed visible changes in teeth condition and discoloration of the enamel (Figure 6).

Medical history also reports that along with the skeletal lesion, the grandmother of the proband was diagnosed with bilateral glaucoma in the first year of life, for which surgery was performed in infancy. At the age of 28 years, after the injury, enucleation surgery for the affected eye was performed. In addition, she developed facial nerve paralysis at the age of 50. Neurological symptoms were also noted in the proband and his mother, who both presented with headaches. The proband’s mother also had symptoms of migraine.

### 2.3. Family 3

Proband (II-1) is a 30-year-old woman from the Bashkirian family. The first fracture of the right femur was revealed shortly after birth, and the second fracture of the same bone was sustained at the age of 1 year. Throughout her life, multiple fractures of the upper and lower limbs (about 20) have been registered. Since the age of 17, she has been treated with bisphosphonates. At the age of 24, a corrective osteotomy of the tibia with osteosynthesis using low-profile palmar locking compression plates (LCPs) was performed. Currently, metal construction of the thighs and lower legs are installed (at the age of 25 and 26 years, respectively), and fractures of the bones of the feet occur regularly. Against the background of a break in bisphosphonate therapy, an increase in bone pain syndrome not related to physical activity is noted. At the age of 27, pamidronate acid therapy was resumed, at which point the severity of the pain syndrome was maximal, with a Z-criterion of the lumbar spine according to densitometry data of −2.4. During therapy, the patient reported a decrease in the severity of the pain syndrome and also an increase in bone mineral density at various times, with the Z-criterion ranging from 0.3 to 1.4. In addition to bisphosphonates, from November 2019 to November 2020, the patient received teriparatide in a standard dosage. After November 2021, there was a break in bisphosphonate therapy due to the coronavirus pandemic. Since the beginning of 2022, the patient has reported a recurrence of the pain syndrome, despite normal bone mineral density levels (Z = 0.8). Pamidronate acid therapy was resumed with a positive therapeutic effect. The dynamics of pain syndrome indicators and bone mineral density levels during the therapy are presented in Figure 7.

A lateral radiograph of the scull performed at the age of 30 years revealed sclerotic lesions involving the frontal, parietal, and temporal bones with the maximum size in the projection of the coronal suture. These findings were confirmed by a CT scan of the skull bones.

An X-ray examination of the thoracolumbar spine revealed a biconcave appearance of the vertebrae and a wedge-shaped deformity of the vertebral bodies at the entire level of the thoracolumbar spine (Figure 2). During bone scintigraphy of the skeletal system in a 30-year-old individual, no metabolically active zones were detected, which are often found in patients with osteogenesis imperfecta, even outside fracture zones and bone calluses.

Physical examination revealed his height to be 150 cm (−2.02 SD). In addition, the second stage thoracolumbar scoliosis, asymmetry of the chest, deformity of the limbs, limitation of movements in the left elbow joint after fractures, shortening of the left lower limb by 2 cm, and abnormal growth of teeth with yellowing of enamel were noted (Figure 6). Examination of both eyes revealed the bluish-gray discoloration of his sclera. A dense bone mass on the head in the projection of the frontoparietal bones was palpated.

The proband’s mother had short stature—149 cm (−2.02 SD) and multiple bone prominences of the calvaria on palpation. According to the medical history, she sustained the first and only fracture of the right femur at the age of 48 years. In addition, she periodically experienced severe pain in different parts of the spine. Patients from this family have never had nerve palsies or other neurological or ophthalmic pathology. 

## 3. Discussion

To date, the clinical and genetic characteristics of 15 patients with CDL aged 6 to 85 years from eight families have been reported in the literature. Three pathogenic heterozygous variants in the *SGMS2* gene were identified in these patients. In six families, the disease was caused by the c.148C>T(p.Arg50Ter) nonsense variant, and in the other two families, the disease was caused by missense variants p.Ile62Ser and Met64Arg, respectively. It is worth noting that Robinsson et al. (2021) reported a family with incomplete penetrance of the *SGMS2* gene [7]. Clinical manifestation of the disease is characterized by a combination of multiple spontaneous spinal and/or peripheral fractures resulting from severe osteoporosis and specific doughnut-shaped sclerotic skull lesions. In addition, the bone biopsy performed in the previous studies of affected patients showed a general decrease in bone volume and low and heterogeneous mineralization of the bone matrix with a chaotic arrangement of collagenous fibrils under polarized light [4,8].

The height of most patients at the time of examination was within the normal range; only the proband and his mother from the third family had reduced growth (−2.02 SD). In all patients, except for the 6-year-old proband from the first family, the presence of single or multiple sclerotic skull lesions was revealed. In the proband from the second family, this feature was taken into account, which, along with the clinical picture resembling imperfect osteogenesis, allowed to diagnose CDL and confirm the diagnosis by direct automatic Sanger sequencing, which identified the pathogenic variant at the “hot spot” of the *SGMS2* gene.

The skull lesions are a fascinating feature of the disease, limited to the vault of the skull. Standard skull X-rays reveal multiple areas of bone involvement, with the appearance of ring-shaped formations resembling a doughnut, lytic lesions of the skull vault, and thickening of the skull vault and/or base. In some cases, sclerotic lesions of the skull bones lead to the protrusion of the outer plate of the skull vault, clinically manifested as a palpable thickening in this area of the head (Figure 4). Skull lesions often progress asymptomatically and are usually detected incidentally on computed tomography (CT) or magnetic resonance imaging (MRI) of the brain. Characteristic skull bone lesions can be diagnosed with standard X-rays, but CT and MRI, by detecting small formations in the diploic space of the skull, allow for early diagnosis of the disease in patients with only multiple fractures in the absence of clinically palpable subcutaneous thickening of the head. It should be noted that doughnut-shaped skull lesions are not observed in early childhood due to age-related structural peculiarities (the diploic space is not well developed in children) and can only be detected at a later age [9]. With age, the number and size of lesion foci increase. In addition, it is important to verify these bone lesions as a benign process and differentiate them from other lytic lesions of the skull vault (intradiploic epidermoid cysts, hemangiomas, eosinophilic granulomas, etc.), including malignant primary and metastatic lesions [1,10,11]. The detection of typical changes in the skull vault bones in combination with the patient’s age, family history, and clinical symptoms of the disease is of fundamental importance for differential diagnosis with osteogenesis imperfecta and for establishing the correct diagnosis.

A recent study by Yan et al. identified the *SGMS2* gene as a regulatory gene of late embryonic craniofacial development in mice; however, the restriction of sclerotic lesions to the skull bones still remains unclear [12]. In 2019, Yoshikawa et al. reported that *SGMS2* knockdown in primary murine osteoblasts reduces the expression of the nuclear factor κB (NF-κB) ligand and increases the expression of osteoprotegerin, thereby reducing osteoclastogenesis [13]. However, these studies do not reflect the situation in people because the nucleotide version in the *SGMS2* gene leads to osteoporosis, heterogeneous mineralization, trabecularization of the cortical plate, and osteolithic/calvarial lesions of the cranial vault.

Effective pathogenetic treatment for the discussed skeletal dysplasia is currently not described. Treatment is symptomatic, aimed at preventing the progression of osteoporosis, and includes the use of bisphosphonate-based medications. Given the limited observations, it is not possible to assess the effectiveness of this treatment. However, in previously described cases, intravenous bisphosphonate therapy was associated with a significant increase in BMD in the lumbar spine and restoration of the shape of previously compression-deformed vertebral bodies, as well as preventing further fractures of the long bones of the skeleton and reducing the intensity of back pain [4,7].

All patients that underwent densitometry were diagnosed with a pronounced decrease in BMD. The proband from the first family was diagnosed with severe osteoporosis, but on the background of course treatment with bisphosphonates which he began to receive at the age of 5 years, positive dynamics in BMD were noted. Obviously, the long-term treatment of osteoporosis and the safety of using bisphosphonates in young patients with nucleotide variants in the *SGMS2* gene have yet to be evaluated. The onset age of fractures ranged from the first days of life to 48 years. Severe progressive scoliosis is not typical for this disease.

It is important to note that most of the affected family members first examined by doctors were children who complained of back pain. During spine radiography, multiple vertebral compression fractures, which were the cause of pain, were found. The identification of other affected members was carried out based on a genealogical analysis and a study of their medical history, which reported that multiple peripheral fractures occurred at different ages. Thus, the proband from the third family was diagnosed with a severe course of the disease with the onset of bone fractures right after birth, severe osteoporosis, back pain, and thoracic vertebrae compression fractures. Therefore, the patient was misdiagnosed with imperfect osteogenesis for a long time, and a small dense formation on the back of the head 1 × 1 cm in size was not associated with the disease until the molecular confirmation of the CDL. At the same time, her mother, from which the disease was inherited, sustained the first fracture of the right femur only at the age of 48 years. In addition, according to her medical history, she periodically experienced severe pain in different parts of the spine, and densitometry revealed signs of severe osteoporosis of the spine. Multiple sclerotic lesions of the skull bones were found on examination. A diffuse pain syndrome was observed in all bones of the proband from the third family, which was not associated with fractures and persisted even after normalization of the level of BMD. The use of bisphosphonates had a positive clinical effect, as previously noted in patients with osteogenesis imperfecta [14].

The analysis of phenotypic manifestations in the patients observed in this study and presented in the literature indicates polymorphism of the clinical manifestations of the CDL in patients with the recurrent variant in the *SGMS2* gene. The presence of significant clinical polymorphism (including intrafamilial) in all of these cases allows us to conclude the possible effect of the genetic background and epigenetic modifiers on the formation of the disease phenotype. In most patients, the first fractures occur in childhood; however, the age of clinical manifestation widely varies. Differences in the height indicators of patients with the presence of multiple fractures were noted. In some patients, height was within the normal range, while in others, the growth was reduced by more than 2 SD. Along with skeleton involvement, some patients have symptoms of damage to the nervous system in the form of migraine, cephalgia, recurring paralysis of the facial, trigeminal, and oculomotor nerves, carpal tunnel syndrome, dystonia, and tremor. However, the severity of skeleton and nervous system involvement in patients with nonsense and two missense variants differed significantly. In two patients with p.Ile62Ser or Met64Arg variants reported in the literature, spontaneous fractures occurred in the neonatal period and led to decreased height as a result of deformations of the spine and limbs. The radiological examination revealed signs of spondylometaphyseal dysplasia. The more pronounced symptoms of nervous system damage were also noted. In addition to transient cranial nerves paralysis, patients were diagnosed with sensory neuropathy, hearing loss, and tendon hyporeflexia. The pathogenesis of CDL has not been fully understood yet, but it is assumed that neurological symptoms are associated with a dysfunction of the gene product, the enzyme sphingomyelin synthase, which is involved in the synthesis of myelin in the structures of the nervous system. Some of the patients examined by us (4/11) had transient recurrent facial nerve paralysis in adulthood, affecting both the upper and lower parts of the face that resolved on their own within a month. Two patients from the second family suffered from headaches since childhood. In addition, the presence of congenital bilateral glaucoma in one of the examined patients was noted. Interestingly, congenital glaucoma was also reported in one patient with CDL in a study by Pekkin M. et al. in 2019 [4]. Undoubtedly, this observation requires further accumulation of clinical data in patients with CDL. It should be noted that in the spectrum of phenotypic manifestations, serious damage to the teeth was found in some of the patients with a severe course of the disease, which led to their early destruction, wearing out of the incisal edges, and yellowing of the enamel, although other patients did not have the signs of teeth involvement. Another additional finding in CDL was a shortening of the fourth–fifth metacarpals previously reported by Jaakkola E. et al. [15]. We were able to detect similar features in the cousin of the proband in the first family. As the disease is quite rare, it is necessary to describe in detail all the phenotypic characteristics of patients with various types of variants in the *SGMS2* gene, which will help to optimize its diagnosis.

## 4. Materials and Methods

A comprehensive examination of 11 patients from three unrelated families with clinical and radiological signs of CDL with bone fragility was carried out. To clarify the diagnosis, following methods were used: genealogical analysis, clinical examination, neurological examination according to the standard technique with an assessment of the psychoemotional sphere, radiography, whole exome and whole genome sequencing, direct automatic Sanger sequencing.

Genomic DNA was extracted from whole venous blood via a Wizard Genomic DNA Purification Kit (Promega, Madison, WI, USA) and QIAamp DNA Blood Mini Kit (QiaGen, Hilden, Germany) following the manufacturer’s protocol.

The concentration of DNA and DNA libraries were measured on a Qubit 2.0 instrument using reagents (Qubit Broad Range, Qubit High Sensitivity assay kits) from the manufacturer according to standard protocol. Concentration and fragment size of each library were checked on a TapeStation 4200 using reagents (high-sensitivity DNA D1000) from the manufacturer according to the standard protocol.

For sample preparation by whole exome sequencing, selective capture of a DNA region was used, designed to encode regions (Illumina TruSeq Exome Kit and IDT xGen Exome Research Panel). Whole exome sequencing was carried out with an average coverage ×60; the number of targeted areas had a coverage ≥×10–99%, uniformity Pct > 0.2*mean—99%.

The proband’s whole genome was sequenced using the MGIEasy stLFR Library Prep Kit, following the manufacturer’s protocol (MGI Tech Co., Ltd., Shenzhen, China). The paired-end sequencing was performed with 2 × 100 bp reads on a DNBSEQ-T7 (MGI Tech Co., Ltd., Shenzhen, China). The sequencing data had an average coverage of at least 35. Sequencing data were processed using a standard pipeline with our in-house software (http://ngs-data.ru/, accessed on 31 October 2022). Sequenced fragments were visualized with Integrative Genomics Viewer (IGV) software (2013–2018 Broad Institute, Cambridge, MA, USA and the Regents of the University of California, Oakland, CA, USA).

The revealed gene modifications were designated in accordance with HGVS nomenclature (https://varnomen.hgvs.org/, accessed on 31 October 2022). The Genome Aggregation Database (gnomAD v.2.1.1) was used to determine the allele frequency of newly discovered variants. The following online prediction programs were utilized to determine pathogenicity in silico: BayesDel_addAF (https://fengbj-laboratory.org/BayesDel/BayesDel.html), DEOGEN2 (http://deogen2.mutaframe.com), EIGEN (https://eigen.tuxfamily.org/index.php?title=Main_Page), FATHMM-MKL (https://fathmm.biocompute.org.uk/), LIST-S2 (https://precomputed.list-s2.msl.ubc.ca/documentation), MutationAssessor (http://mutationassessor.org/r3/), MutationTaster (http://www.mutationtaster.org/), PrimateAI, SIFT/Provean (http://provean.jcvi.org/index.php), BadMut (http://score.generesearch.ru/services/badmut/), and SpliceAI (https://spliceailookup.broadinstitute.org/), accessed on 31 October 2022. The variants’ clinical significance was evaluated according to the guidelines for massive parallel sequencing (MPS) data interpretation [16].

Validation of the identified variant in probands of the first and third families and diagnosis confirmation in the proband of the second family, as well as genotyping of siblings and parents, were carried out by automatic Sanger sequencing using ABIPrism 3500xl Genetic Analyzer (Applied Biosystems, Foster City, CA, USA) according to the manufacturer’s protocol. The primer sequences used in the study were the following: SGMS2-ex3F: TGAAGACTAGGGGACAATGGA, SGMS2-ex3R: AGCCACTGGGTGATCCATAA. Primer sequences were chosen according to the *SGMS2* (NM_152621.6) reference sequence.

The biochemical parameters calcium, phosphorus, and alkaline phosphase were studied using an automatic analyzer CA-800 (Furuno Electric Co., Ltd., Nagasakishi, Japan) using the manufacturer’s reagents. Serum vitamin D levels were measured by ELISA kit (BioVendor, Brno, Czech Republic).

## 5. Conclusions

Thus, CDL is a rare autosomal dominant disease, which is one of the causes of spontaneous fractures in children. The disease has specific clinical and radiological manifestations, which make it possible to differentiate it from a common hereditary disease characterized by spontaneous fractures as a result of low bone mineral density—osteogenesis imperfecta. The literature data and results of our study allow us to conclude that there is a recurrent variant in the *SGMS2* gene responsible for the occurrence of CDL. The primary analysis of this variant will contribute to optimal molecular genetic diagnostics, which can reduce diagnostic costs and time.

## Figures and Tables

**Figure 1 ijms-24-08021-f001:**
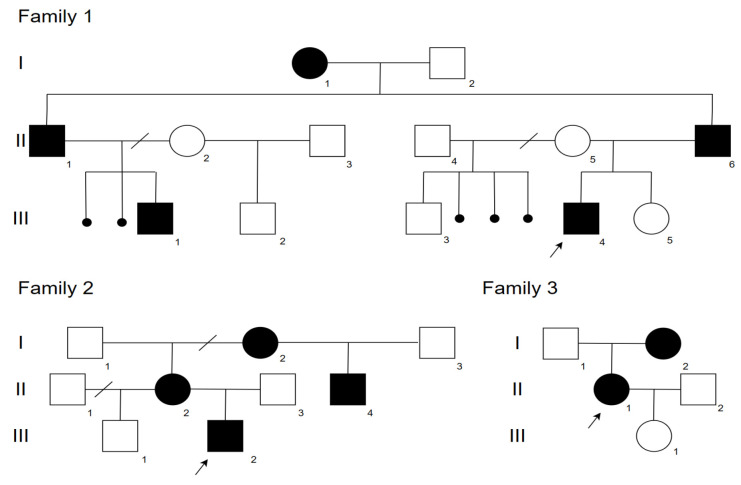
Pedigree charts of the families with a nonsense variant in the *SGMS2* gene: c.148C>T (p.Arg50Ter). Black—affected family members, white—healthy family members, I–III—generations, 1–6—number of the person in the generation.

**Figure 2 ijms-24-08021-f002:**
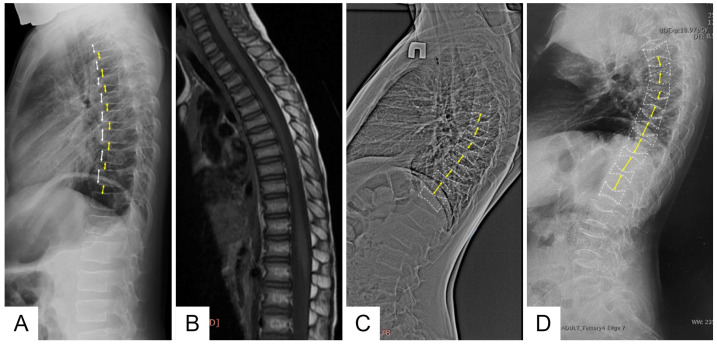
Lateral views of the thoracic and upper lumbar spine of the patients of different ages: radiograph of the 6 y.o. boy (F1 III-4) (**A**)—decreased height and anterior wedging of the vertebral bodies (white arrows), increased height of intervertebral discs (black arrows); T1-weighted MRI of the same patient of the 4 y.o. (**B**); radiographs of the 12 y.o (F2 II-4) (**C**) and 30 y.o. (F3 II) (**D**) patients—severe platyspondyly with accentuated biconcave shape of the vertebral bodies (white dotted lines) and increased height biconvex intervertebral disks (black arrows) more prominent at the thoracic level.

**Figure 3 ijms-24-08021-f003:**
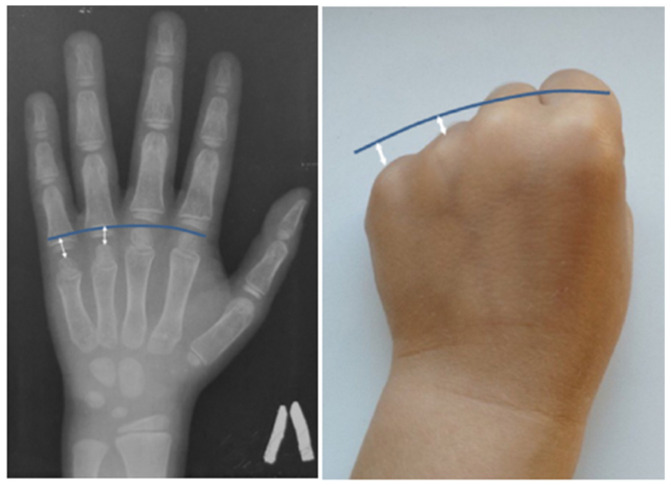
Radiograph and appearance of the hand of the 7 y.o. patient (F1 III-1) with brachymetacarpia of the fourth and fifth digits: relative shortening (white arrows) of the fourth and fifth metacarpals to compare with the metacarpal heads line (blue line).

**Figure 4 ijms-24-08021-f004:**
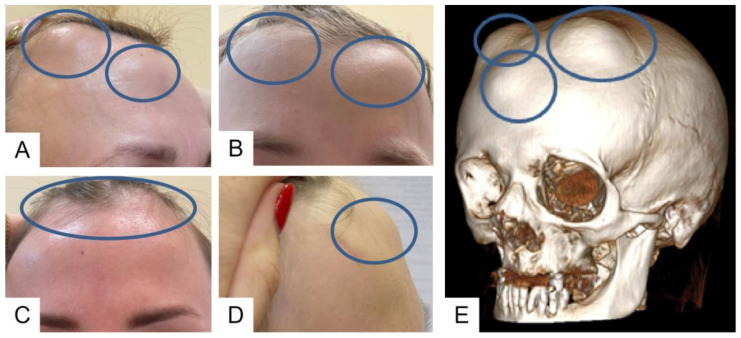
Clinical (**A**–**D**) and CT (**E**) view of the bony prominences of the calvaria of the members of one family.

**Figure 5 ijms-24-08021-f005:**
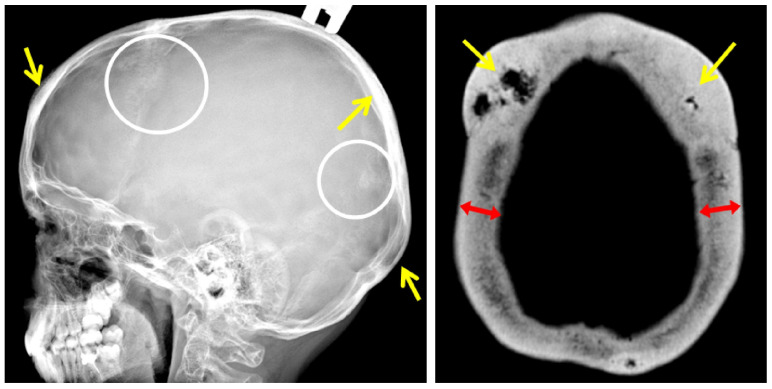
Lateral radiograph of the skull (F2 III-2): sclerotic lesions (white circles) and thickening (yellow arrows) of the bones. Axial CT scan of the skull vault (F2 I-2): doughnut-shaped lesions (yellow arrows) and significant thickening of the cranial bones (red arrows).

**Figure 6 ijms-24-08021-f006:**
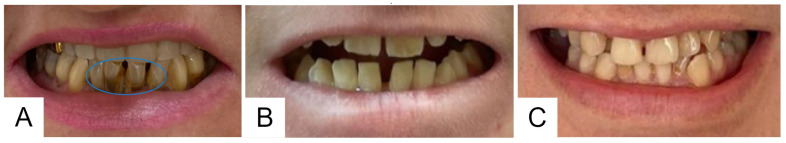
(**A**) (F2 I-2): Lower incisors are discolored, incisal edges are worn from 1 to 3 mm. Clinical detachment lost, tremas, abfraction in the cervical area of the crown, pigmented dentin are noted, and gingival recession is from 2 to 5 mm. Pigmented dentin is determined (blue oval). (**B**) (F2 II-4) and (**C**) (F3 II): abnormal growth of teeth and yellowing of the enamel.

**Figure 7 ijms-24-08021-f007:**
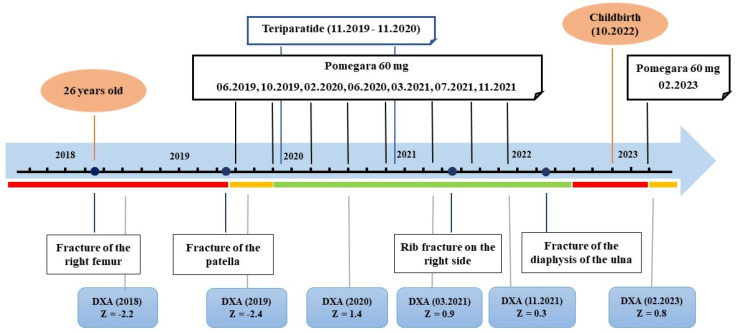
The dynamics of BMD, pain syndrome, and fractures in patient II-1 on the background of conducted therapy. The red line represents pain syndrome of more than six points on the visual analog scale, the yellow line represents pain syndrome from two to five points, and the green line represents fewer than two points: DXA, dual-energy X-ray absorptiometry, Z-criterion of mineral bone density of the lumbar spine.

**Table 1 ijms-24-08021-t001:** Clinical findings in the 11 subjects with a nonsense variant in the *SGMS2* gene: c.148C>T (p.Arg50Ter).

	Family 1					Family 2				Family 3	
Patient	III-4	III-1	II-6	II-1	I-1	III-2	II-2	II-4	I-2	II-1	I-2
Sex	m	m	m	m	f	f	f	m	f	f	f
Age (years)	6	7	33	35	56	11	31	11	52	31	53
Height (SD)	−1.2	−1.8	−0.05	−0.47	−0.34	−0.82	−0.18	−0.05	−0.25	−2.02	−2.02
CDL (number)	-	3	mpl	mpl	mpl	mpl	mpl	2	3	mpl	mpl
BMD (L1–L4) (g/cm^2^)	0.289 (4 y.), 0.357 (6 y.)	0.344	0.887	ND	0.804	0.327	0.708	0.310	0.747	0.084	0.442
Z-score (L1–L4); T-score (>50 y.) (L1–L4);	−3.2; L2-4 (4 y.) −4.2; L2-4.5 (5 y.) −1.9 (6 y.)	−2.6; L2-3.4	−1.6; L1-2.1 TB-2.4	ND	−2.2; L1-3.5	−3.7; L1-4.3	−3.1; L1-3.9	−3.9; L1-4.5	−2.8; L1-3.4	−2.4; (20 y.)	−6.1; L3-7.1
Age of first fracture (years)	1.7	-	10	16	5	1.6	6	5	5	0.1	48
Number of peripheric fractures	-	-	40	20	13	4	6	7	10	20	1
Scoliosis (stage)	-	-	-	1	1	1	1	2	2	2	1
Age of treatment initiation by BP (year)	5	-	-	-	-	11	-	11	-	17	-
Facial nerve paralysis	-	-	4 times	2 times	1 time	-	-	-	1 time	-	-

Mpl, multiple; BMD, bone marrow density; ND, no data; BP, bisphosphonates.

## Data Availability

Not applicable.

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
