# Peer review of "Clinical and Genetic Characteristics of Calvarial Doughnut Lesions with Bone Fragility in Three Families with a Reccurent SGMS2 Gene Variant"

_ijms, 2023, doi:10.3390/ijms24098021_

Round 1

Reviewer 1 Report

The authors describe the clinical and genetic characteristics of  a rare skeletal disorder which is caused by mutations in the SGMS2 gene  encoding for the enzyme sphingomyelin synthase 2. In this study, 11 patients from 3 unrelated families are characterized and a recurrent gene variant in SGMS2 was identified. Even though this manuscript does not provide novel mechanistic insights it extends the phenotypic spectrum of a rare disease. 

The manuscript is well written and easy to read. The authors mention in the discussion that there is certain risk to misdiagnose these patients with osteogenesis imperfecta (L291ff). Similarities with and differences to OI could be discussed in more detail to prevent this risk of misdiagnosis.    

Very minor things/typos:

L8: the affiliation should be revised

L53: SF or SM?

Table: BMD g/sm2?, all patients in bold

Author Response

Dear Reviewer,

The authors would like to express their gratitude for your careful review of the manuscript and the insightful comments provided. We believe that taking into account your suggestions has significantly improved the quality of the manuscript. In the revised version, we have made every effort to thoroughly check the language and spelling. Following your esteemed recommendations, the authors have made changes and additions to the "Discussion" section to prevent the risk of misdiagnosis of osteogenesis imperfecta.

The skull lesions are a fascinating feature of the disease, limited to the vault of the skull. Standard skull X-rays reveal multiple areas of bone involvement, with the appearance of ring-shaped formations resembling a donut, lytic lesions of the skull vault, thickening of the skull vault and/or base. In some cases, sclerotic lesions of the skull bones lead to the protrusion of the outer plate of the skull vault, clinically manifested as palpable thickening in this area of the head (Fig. 4). Skull lesions often progress asymptomatically and are usually detected incidentally on computed tomography (CT) or magnetic resonance imaging (MRI) of the brain. Characteristic skull bone lesions can be diagnosed with standard X-rays, but CT and MRI, by detecting small formations in the diploic space of the skull, allow for early diagnosis of the disease in patients with only multiple fractures in the absence of clinically palpable subcutaneous thickening of the head. It should be noted that donut-shaped skull lesions are not observed in early childhood due to age-related structural peculiarities (the diploic space is not well developed in children) and can only be detected at a later age [9]. With age, the number and size of lesion foci increase. In addition, it is important to verify these bone lesions as a benign process and differentiate them from other lytic lesions of the skull vault (intradiploic epidermoid cysts, hemangiomas, eosinophilic granulomas, etc.), including malignant primary and metastatic lesions [1, 10, 11]. The detection of typical changes in the skull vault bones in combination with the patient's age, family history, and clinical symptoms of the disease is of fundamental importance for differential diagnosis with osteogenesis imperfecta and for establishing the correct diagnosis.

We appreciate your helpful comment and have corrected the typos accordingly.

  1. L8: The affiliations have been revised.
  2. L53: SM (sphingomyelin) has been corrected.
  3. Table: BMD has been corrected to g/cm2, and all patients are now in bold.

Overall, we are grateful for your kind review and constructive comments.

Sincerely,

The collective of authors.

Reviewer 2 Report

The case report submitted by Merkuryeva Elena et al. addressed the clinical and genetic characteristics of patients with Calvarial doughnut lesions (CDL) with bone fragility caused by a pathogenic nonsense variant in the SGMS2 gene: c.148C>T (p.Arg50Ter). Authors have shown that the wide inter-familial and intra-familial phenotypic variability in patients with a detected recurrent variant in the SGMS2 gene. And claimed that the recurrent variant in the SGMS2 gene must be taken into consideration in the diagnosis of the disease. The relevance of the report was well highlighted and described, and no breaches in ethical practice were noted. This case report is well written, and it adds to current knowledge about bone fragility disorders along with osteogenesis imperfecta. But the report missed some important information which needs to be added to the present report as mentioned below.

Major Concern:  

The authors describe that the protein product of the SGMS2  gene is sphingomyelin synthase 2 which involves bone mineralization. Defective enzyme activity leads to clinical manifestations of the disease in patients. But Authors have not shown any relevant data in the present case report. Did the authors estimate the sphingomyelin synthase 2 levels in patient samples?

Please correct minor mistakes in the Figure 5 legend, Lines 177, 178, and 221. It’s Skull instead of “Scull”.   

Authors also need to explain all patient’s outcomes and follow up after starting medication or any treatments prescribed to patients.

Please add the missing reference of Jaakkola et al. on line 337 in the discussion section. The reference list is too short. Please add the necessary references in the discussion section.

Please describe detailed methods and kit/reagent information used for evaluating the calcium phosphate metabolism (Calcium, ALP, Vit D, and inorganic phosphorus). 

Author Response

Dear Reviewer,

We would like to express our gratitude for your insightful analysis of our manuscript, as well as for your comments and suggestions, which have significantly improved the quality of our data analysis and presentation. In the revised version of this work, we have made every effort to take into account the remarks and suggestions presented in your review.

Please find below our responses to your comments.

  1. The authors describe that the protein product of the SGMS2  gene is sphingomyelin synthase 2 which involves bone mineralization. Defective enzyme activity leads to clinical manifestations of the disease in patients. But Authors have not shown any relevant data in the present case report. Did the authors estimate the sphingomyelin synthase 2 levels in patient samples?

Regarding your comment on the measurement of sphingomyelin synthase 2 activity, we would like to clarify that this parameter is not included in the clinical examination protocol for patients in our country. Furthermore, according to a previous study (Pekkinen et al, 2019), the synthesis rate of sphingomyelin and gene expression did not differ from normal levels in blood and fibroblasts. Therefore, the expression in patient samples was not measured, as it has been previously shown to be within normal range.

  1. Please correct minor mistakes in the Figure 5 legend, Lines 177, 178, and 221. It’s Skull instead of “Scull”.   

We have rectified the typographical error as per your suggestion.

  1. Authors also need to explain all patient’s outcomes and follow up after starting medication or any treatments prescribed to patients.

We have carefully reviewed your recommendation and have made additional clarifications to the «Results» section:

Family 1: During the patient's observation, there was a tendency towards an increase in the pain syndrome in the back area (pain occurred during coughing, sneezing, and sudden movements), as well as a decrease in BMD values in the lumbar spine according to densitometry data (BMD=0.259 g/cm2, Z-score: -4.2 SD), which served as a medical indication for initiating treatment with bisphosphonate-based medications (pamidronate acid at a dose of 1 mg/kg/day for 3 consecutive days, courses every 4 months). After 2 courses of treatment, a positive trend was observed: back pain was relieved, and BMD values increased (Table 1).

Family 2: At the age of 11 years, therapy with bisphosphonates was initiated. Less than 6 months have elapsed since the initiation of treatment, and at present, it is difficult to assess the efficacy of the treatment.

We have also made appropriate modifications to the «Discussion» section as per your suggestions:

Effective pathogenetic treatment for the discussed skeletal dysplasia is currently not described. Treatment is symptomatic, aimed at preventing the progression of osteoporosis, and includes the use of bisphosphonate-based medications. Given the limited observations, it is not possible to assess the effectiveness of this treatment. However, in previously described cases, intravenous bisphosphonate therapy was associated with a significant increase in BMD in the lumbar spine and restoration of the shape of previously compression-deformed vertebral bodies, as well as preventing further fractures of the long bones of the skeleton and reducing the intensity of back pain [4,7].

  1. Please add the missing reference of Jaakkola et al. on line 337 in the discussion section. The reference list is too short. Please add the necessary references in the discussion section.

Specifically, we have added the missing reference of Jaakkola et al. on line 374 in the discussion section. We also agree that the reference list was too short and have added the necessary references in the discussion section.

Furthermore, we have thoroughly reviewed all the references and made changes to the "References" section (items 10 and 11) to ensure accuracy and completeness.

  1. Please describe detailed methods and kit/reagent information used for evaluating the calcium phosphate metabolism (Calcium, ALP, Vit D, and inorganic phosphorus). 

We would like to express our gratitude for your valuable comment. We have incorporated additional clarifications in the «Materials and Methods» section of our manuscript.

The biochemical parameters - calcium, phosphorus, alkaline phosphase were studied using an automatic analyzer CA-800 (Furuno Electric Co. Ltd., Japan), using the manufacturer's reagents. Serum vitamin D levels were measured by ELISA kit (BioVendor, Czech Republic).

Overall, we are grateful for your kind review and constructive comments.

Sincerely,

The collective of authors.

Reviewer 3 Report

In the paper "Clinical and genetic characteristics  of calvarial doughnut lesions with bone fragility in three families with a reccurent SGMS2 gene variant" the authors presented clinical features of 11 patients from three different families diagnosed with Calvarial doughnut lesions (CHL). 

Current knowledge on the CHL is clearly presented in the introduction, conducted M&M are described in details and obtained results are clearly presented and properly discussed.

However, there are some minor issues related to the quality of figures which should be resolved prior publication:

Figure 2: I would suggest to increase the size of white/black arrows and/or change their color because their visibility is currently not optimal. I would also suggest to put all parts of the figure (A-D) into one row instead of two.  

Figures 2, 4 and 6: the letters in white boxes should be centered in the boxes and the font size should be uniform in all figures. I would suggest to put white boxes right in the corner and remove any space between box and figure edge (currently they are several milimeters from the figure edge)

Figure 5: I would suggest to change the color of arrows to improve their visibility

The present paper is important because up to date there are only 15 patients from 8 families with CHL described in literature and this paper will contribute to better understanding of CHL. 

Author Response

Dear Reviewer,

The authors would like to express their gratitude for your careful review of the manuscript and the valuable feedback provided. We believe that incorporating your suggestions has significantly improved the quality of the manuscript. In accordance with your recommendations, the authors have summarized the main points and made changes to the figures to enhance their visibility.

  1. Figure 2: I would suggest to increase the size of white/black arrows and/or change their color because their visibility is currently not optimal. I would also suggest to put all parts of the figure (A-D) into one row instead of two.  

Figure 2: We changed the color of the arrows to optimize their visibility. Additionally, we have placed all parts of the figure (A-D) in one row instead of two.

  1. Figures 2, 4, and 6: The letters in the white boxes have been centered and the font size has been made uniform across all figures.

We have also placed the white boxes in the corner and removed any space between the box and the figure edge, which was previously several millimeters.

  1. Figure 5: We have changed the color of the arrows to improve their visibility, as per your recommendation.

Overall, we are grateful for your kind review and constructive comments.

Sincerely,

The collective of authors.